# Numerical Simulation Study of Multi-Field Coupling for Laser Cladding of Shaft Parts

**DOI:** 10.3390/mi14020493

**Published:** 2023-02-20

**Authors:** Changlong Zhao, Chen Ma, Junbao Yang, Ming Li, Qinxiang Zhao, Hongnan Ma, Xiaoyu Jia

**Affiliations:** College of Mechanical and Vehicle Engineering, Changchun University, Changchun 130022, China

**Keywords:** laser cladding, multi-field coupling, numerical simulation, residual stress

## Abstract

Since shaft parts operate under harsh environments for a long time, many critical parts suffer from corrosion, wear and other problems, leading to part failure and inability to continue in service. It is imperative to repair failed parts and increase their service life. An orthogonal experimental scheme is designed to numerically simulate the process of laser cladding of Inconel 718 alloy powder on 4140 alloy structural steel based on the ANSYS simulation platform, derive the relationship equation of cladding layer thickness according to the heat balance principle, establish a finite element model, couple three modules of temperature field, stress field and fluid field, and analyze different modules to realize the monitoring of different processes of laser cladding. The optimal cladding parameters were laser power 1000 W, scanning speed 15 rad/s, spot radius 1.5 mm, thermal stress maximum value of 696 Mpa, residual stress minimum value of 281 Mpa, and the degree of influence of three factors on thermal stress maximum value: laser power > spot radius > scanning speed. The pool in the melting process appears to melt the “sharp corner” phenomenon, the internal shows a double vortex effect, with a maximum flow rate of 0.02 m/s. The solidification process shows a different shape at each stage due to the different driving forces. In this paper, multi-field-coupled numerical simulations of the laser cladding process were performed to obtain optimal cladding parameters with low residual stresses in the clad layer. The melt pool grows and expands gradually during melting, but the laser loading time is limited, and the size and shape of the melt pool are eventually fixed, and there is a vortex flowing from the center to both sides of the cross-section inside the melt pool, forming a double vortex effect. The solidification is divided into four stages to complete the transformation of the liquid phase of the melt pool to the solid phase, and the cladding layer is formed. The multi-field-coupled numerical simulation technique is used to analyze the temperature, stress and fluid fields to provide a theoretical basis for the residual stress and surface quality of the clad layer for subsequent laser cladding experiments.

## 1. Introduction

The 4140 alloy structural steel is a medium carbon steel with high strength and hardenability, good toughness, low thermal deformation during quenching, high creep strength and lasting strength at high temperature, which is usually used in milling machine spindles, steel couplings, drums, etc., and in oil deep well drill pipe joints and salvage tools, etc. [1,2,3]. However, the working environment is usually harsh, and the steel surface is subject to wear, cracks and other defects. The common repair methods in the market are laser cladding [4], brush plating [5], spraying process [6], etc. However, the brush plating coating is shallow, the strength of the bond is low, there is a large residual stress generated, and the coating easily falls off [7]; the spraying process itself is low strength, not up to the strength of the metal itself, and the spraying layer will produce certain porosity, flow hanging, bubbles, unevenness and other problems [8,9]. However, laser cladding is a directional energy deposition (DED) technology [10,11], the principle of which is to use a high-energy density laser beam to metallurgically bond the powder to the substrate, with fast cooling and fast heating, which can improve the surface properties of the substrate material, making it stronger, have better wear resistance, higher thermal fatigue strength, etc. [12,13,14]. It is commonly used in additive manufacturing for repairing failed parts, strengthening surfaces, etc. [15,16].

Numerous numerical simulation models of laser cladding have been established and analyzed by domestic and foreign scholars [17,18,19,20]. Zhao Shengju et al. conducted a numerical simulation of laser cladding of TC4 alloy and showed that the laser power and scanning speed were the main influences on the temperature field distribution [21]. Li Jinhua et al. conducted a laser cladding simulation study of H13 steel based on the COMSOL simulation platform, and the results showed that the peak point of the von Mises thermal stress cycle curve has an important effect on the melt pool depth [22]. Xu Mingsan et al. established a three-dimensional numerical model of the coupled powder–gas–light temperature field, and the study showed that the laser power is the main factor in the change of the temperature field amplitude during the powder feeding process of laser cladding, and the powder carrier is the main factor affecting the powder temperature field distribution, both of which jointly affect the change of the temperature field during the melting process [23]. Xing Han et al. established a numerical model of pulsed laser cladding with a disc laser, which considered the interaction between the powder flow and the pulsed laser beam as well as the effects of surface tension and buoyancy on the fluid flow in the melt pool, and the results showed that pulsed laser cladding facilitates the formation of fine grain structures and obtains better-quality clad layers [24]. The above scholars showed that in the process of building the model, only for the numerical simulation of flat parts, the establishment of a finite element model is relatively convenient and simple. In this paper, a mathematical model of the wear location on the part surface of 4140 alloy structural steel used as a shaft part is established to realize the numerical simulation process of laser cladding on the circular arc surface.

The current numerical simulation of laser cladding is mainly limited to a single module or two modules for modeling. Liu Wei-Wei et al. combined the temperature field with the stress field to investigate the effect of three different scanning methods of laser cladding on the deformation of the parts, and the results showed that the same direction of the cladding layer can make the temperature field more uniform and reduce the deformation of the parts [25]. Liu Huaming et al. developed a finite element model of the temperature and stress fields of single-pass laser cladding, and the results showed that the laser power and scanning speed play a major role in the variation of the temperature and stress fields [26]. Ren Zhonghe et al. established a laser cladding temperature field model and analyzed the microstructure of the clad layer at different depths. The results showed that the addition of CeO_2_/TiO_2_ to the powder could improve the nucleation rate of the clad layer and obtain a uniform and fine microstructure [27]. The above scholars only analyzed the temperature and stress in the laser melting process but failed to go deep into the interior of the melt pool, did not reveal the melting and solidification process of the melt pool, and could not analyze the laser cladding process from multiple angles.

In this paper, we use a multi-field coupled numerical simulation technique to couple the temperature field, stress field and fluid field to analyze the internal changes of the melt pool during the cladding process and the results after the cladding. Temperature field extraction thermal cycle curves, contour plot of the temperature distribution, etc., were used to study the effect of different cladding parameters on the cladding layer size. The stress field is extracted from the thermal stress maximum, stress distribution cloud, residual stress distribution curve, etc. The degree of influence of laser power, scanning speed and spot radius on the thermal stress maximum is analyzed by using extreme difference and analysis of variance, and the optimal cladding parameters and minimum residual stress are obtained. The fluid field is analyzed from three aspects: melt pool melting, melt pool solidification and internal flow velocity of the melt pool, extracting melt pool melting, solidification and velocity vector clouds, studying the melt pool change process, realizing the monitoring of melt pool state change, and providing theoretical basis for subsequent laser cladding experiments.

## 2. Numerical Simulation

### 2.1. Selection of Experimental Materials

The elemental composition of both substrate and powder were derived from the material library of JMatPro software. The 4140 alloy structural steel was selected as the workpiece for the numerical simulation of laser cladding, and the elemental composition is shown in Table 1, with Fe as the remaining component. Inconel 718 alloy powder was selected for the cladding powder, and the elemental composition is shown in Table 2, with Ni as the remaining component.

Based on JMatPro software, the thermophysical parameters of the matrix and powder with temperature were calculated using CALPHAD method, and the calculated results were exported and plotted graphically to obtain the thermophysical properties as shown in Figure 1.

From the thermophysical curve of matrix and powder with temperature, it can be seen that 4140 alloy structural steel starts to change from solid phase to liquid phase at 1432 °C, which is the solid phase point, and completely changes to liquid at 1496 °C, which is the liquid phase point, and the solid–liquid coexistence zone is between the two temperature points. Inconel 718 alloy powder has a solid phase point of 1220 °C and a liquid phase point of 1360 °C. The Young’s modulus, density and thermal conductivity of 4140 alloy structural steel and Inconel 718 alloy powder drop sharply at the solid phase point, and as the temperature continues to rise, the Young’s modulus stabilizes when the liquid phase point is reached, the density continues to drop, and the thermal conductivity increases as the temperature rises. The specific heat capacity, Poisson’s ratio, average expansion coefficient, linear expansion coefficient and enthalpy parameters increase sharply after reaching the solid phase point with the increasing temperature, and the average expansion coefficient, linear expansion coefficient and enthalpy parameters continue to increase after reaching the liquid phase point; the specific heat capacity increases sharply between the solid phase point and the liquid phase point, then decreases sharply and finally stabilizes. Poisson’s ratio stabilizes after reaching the liquid phase point.

### 2.2. Build Finite Element Model

Modeling assumptions for laser cladding processes.

(1)The laser energy has a constant Gaussian distribution.(2)The material properties are assumed to be isotropic.(3)The concentration of the powder stream ejected from the powder nozzle is Gaussian distributed, and the powder entering the melt pool melts instantaneously.

In order to improve the accuracy and speed of the numerical simulation, the model needs to be simplified. Since most of the shaft parts are axisymmetric models, a 1/2-axis part model was created for the numerical simulation of laser cladding. The modeling was performed using 3D software and imported into the ANSYS Workbench simulation platform. The meshing was performed using Solid70 cells with a resolution of five. The different parts of the model were named as follows: RFC for the entire cladding model, A1 for the cladding surface and A2 for the cladding and substrate bonding surface. The results of naming and meshing of each part are shown in Figure 2.

### 2.3. Building a Theoretical Model

The choice of the heat source has a great influence on the accuracy of the numerical simulation. The two-dimensional heat source ignores the longitudinal heat flow and restricts the heat source to the X and Y planes. Compared with the two-dimensional heat source, the three-dimensional heat source takes into account the energy absorption in the depth direction and the existence of longitudinal heat flow; thus, the numerical simulation results are more accurate [28]. In this paper, the Gauss three-dimensional body heat source is selected to simulate the laser melting process with the distribution function.
(1)$$q(x,y,t)=\frac{Q}{\pi R_{0}^{2}}\exp\{-\frac{2[{\left(x-x_{0}\right)}^{2}+{\left(y-v_{0}t\right)}^{2}]}{R_{0}^{2}}\}$$
where q(x,y,t) is the heat flux at the lower (x,y) positions at time t, Q is the laser power, $R_{0}$ is the laser beam radius, $x_{0}$ is the coordinate of the laser center in the x direction, and $v_{0}$ is the laser scanning speed.

Many scholars model the temperature field of laser cladding according to empirical values, fixing the size and shape of the cladding layer, and the model cannot meet the numerical simulation under different cladding parameters [29,30]. In this paper, a simplified equation for the geometric size of the cladding layer is derived from the thermal equilibrium principle, and the calculation of the cladding layer height is carried out for the temperature field modeling, which provides a basis for the optimization of the process parameters in terms of temperature and cooling rate.

The effective utilization of laser heat is
(2)$$\beta =\frac{\varepsilon V_{f}}{P}\left(k_{f}+k_{j}\frac{\eta }{1-\eta }\right)$$
(3)$$\varepsilon =\frac{V_{f}^{\prime }}{V_{f}}$$
where ε is the powder factor into the melt pool, $V_{f}^{\prime }$ is the amount of powder into the cladding layer, $V_{f}$ is the powder feeding rate, P is the laser power, $k_{f}$ and $k_{j}$ are the material physical parameters related to the heating temperature after the powder melting, and η is the dilution rate.

The expression for the thickness of the cladding layer is
(4)$$S_{f}=\frac{3}{2}\times \frac{1-\eta }{\rho_{f}\left[\left(1-\eta \right)k_{f}+\eta k_{j}\right]}+\frac{\beta P}{V_{g}D}$$
where $\rho_{f}$ is the density of the powder, $V_{g}$ is the scanning speed, and D is the spot diameter.

Combining Equations (2)–(4) yields
(5)$$S_{f}=\frac{3}{2}\times \frac{\varepsilon V_{f}}{\rho_{f}D}$$

A simplified equation of the relationship between the geometry of the clad layer and the process parameters is derived by the heat balance principle, and combined with the theoretical calculation, the thickness of the clad layer in this paper is 2 mm.

### 2.4. Experimental Protocol Setting

In the laser cladding process, the main parameters affecting the cladding layer are laser power, scanning speed, spot radius, pulse width, powder feed rate and powder loss rate, etc. Among many factors, laser power, scanning speed and spot radius are the main influencing factors [31]. Therefore, this paper designs a three-factor experimental scheme about laser power, scanning speed and spot radius. According to the preliminary preparation experiments and reference literature [22,32], the laser powers are 600, 800, 1000, 1200, 1400 W, scanning speeds are 5, 10, 15, 20, 25 rad/s, and spot radii are 0.5, 1, 1.5, 2, 2.5 mm for each of the five groups of parameters, according to the simulation results of the temperature field under different processes for parameter selection. The experimental scheme is shown in Table 3. The relationship between the internal state of the molten pool and the residual stresses in the molten layer was investigated by numerical simulation of the molten pool geometry, depth and molten pool flow direction using ANSYS Fluent software.

## 3. Analysis of Results

### 3.1. Simulation Results of Temperature Field under Different Process Parameters

Through numerical simulations, the effects of various process conditions on the temperature field of laser cladding were explored to reveal the changes in the temperature field under different process conditions, thus laying the foundation for the design and optimization of the laser cladding process. The simulation results of the temperature field under different process parameters are shown in Figure 3, and the peak temperatures are shown in Table 4.

As can be seen from Table 4, laser power and peak temperature are positively correlated. As the laser power continues to increase, the amount of heat that can be provided per unit time will also increase, which in turn increases the peak temperature of laser cladding, while the scanning speed, spot radius and peak temperature are negatively correlated. As the scanning speed and spot radius continue to increase the area of the heat source applied per unit time, the smaller the heat per unit area, resulting in laser cladding, and the lower the peak temperature of the process. From Figure 3, it can be seen that the heat-affected zone during the melting process is approximately the same by changing only the laser power; the heat-affected zone during the melting process increases with the increase in scanning speed by changing only the scanning speed; the heat-affected zone during the melting process increases with the increase in spot radius by changing only the spot radius. Combining Table 4 with (1) in Figure 3a–c, it can be seen that when the laser power is 600, 800 W, scanning speed is 20, 25 rad/s, spot radius is 2, 2.5 mm, and the cladding temperature is lower than the melting point of the substrate and powder, which fails to meet the cladding conditions. Thus, the laser power is chosen to be 1000, 1200, 1400 W, scanning speed is 5, 10, 15 rad/s, and spot radius is 0.5, 1, 1.5 mm for the design orthogonal experimental scheme. The 1–9 experimental groups are shown in Table 5.

### 3.2. Analysis of Temperature Field Results

In order to investigate the temperature field variation pattern during laser cladding of Inconel 718 powder, temperature extraction was performed at the same locations for different conditions of the laser cladding process. Figure 4 shows the sampling points and markings of the two models in the *X*-axis direction and *Y*-axis direction.

Observing the A2 surface, the melting point of the 4140 alloy structural steel is 1432 °C, and the melting point of the Inconel 718 alloy powder is 1220 °C. If the temperature at the metallurgical bond is higher than the melting point of the substrate, it indicates that the laser cladding process can be realized. According to the orthogonal experimental scheme shown in Table 5, the numerical simulation of laser cladding was carried out, and the data of the *X*-axis and *Y*-axis sampling points and the cross-section of temperature field at the same position were extracted from experimental groups 1–9, and the thermal cycle curves and the cross-sectional morphology of the cladding layer at different sampling points were obtained, as shown in Figure 5, Figure 6 and Figure 7. Each figure (1)–(9) corresponds to experimental groups 1–9 in the orthogonal experimental protocol table.

From Figure 5, Figure 6 and Figure 7, it can be seen that the temperature of the X2 sampling point of experimental group 1, 3, 4, 5, 6, 7, 8 and 9 is higher than the melting point of the base material, and the temperature of the X3 sampling point is lower than the melting point of the base material, and the metallurgical bonding area is judged to be between X2 and X3 sampling points, i.e., the length is 0.5–1 mm, and the width of the molten layer is further obtained between 1–2 mm. The temperature of the X1 sampling point of experimental group 2 is higher than the melting point of the base material, and the temperature of the X2 sampling point is lower than the melting point of the base material. The temperature of the X1 sampling point is higher than the melting point of the base material, X2 sampling point is lower than the melting point of the base material, and the metallurgical bonding area is judged to be between the X1 and X2 sampling points, i.e., the length is 0–0.5 mm, and the width of the molten layer is further obtained between 0 and 1 mm. The model was analyzed by the temperature change curve of each sampling point in the *Y*-axis direction. Y1 and Y4 sampling points of experimental groups 1 and 4 were lower than the melting point of the base material, and Y2 and Y3 sampling points were higher than the melting point of the base material, and the metallurgical bonding area was judged to be between the Y1 and Y4 sampling points, that is, the height of the molten layer was between 0.5 and 1 mm, the depth was between 0 and 0.5 mm, and the longitudinal molten area was further obtained to be between 0.5 and 1.5 mm. The sampling points of Y2 and Y4 of the experimental groups 2, 3, 5, 6 and 9 are lower than the melting point of the base material, and the sampling points of Y3 are higher than the melting point of the base material. It is judged that the metallurgical bonding area is between Y2 and Y4, i.e., the height of the molten layer is between 0 and 0.5 mm, the depth is between 0 and 0.5 mm, and the longitudinal molten area is further obtained between 0 and 1 mm. The sampling points of Y1 and Y5 of experimental groups 7 and 8 are lower than the melting point of the base material, and the sampling points of Y2 and Y4 are higher than the melting point of the base material. It is judged that the longitudinal melting area is between the sampling points of Y1 and Y5, that is, the height of the melting layer is between 1 and 1.5 mm, and the depth is between 0.5 and 1 mm. It is further obtained that the longitudinal melting area is between 1.5 and 2.5 mm.

The sampling points in both *X* and *Y*-axis directions are not rising from the initial temperature of 20 °C because of the heat accumulation during the laser cladding process, which causes the starting temperature of the sampling points to be high. The temperature of the sampling point increases sharply when the heat source passes by and decreases sharply when the heat source is far away, until it reaches the average temperature of the model as a whole, and then, its cooling rate decreases sharply because the radiation effect of the back end of the heat source on the substrate and the cladding layer cause the temperature to drop slowly, and finally, the overall temperature of the model tends to be uniform. With the gradual increase in laser power, the highest temperature of the peak temperature curve gradually increases because the heat flux per unit area increases, resulting in a rise in peak temperature. With the increasing scanning speed, the peak temperature curve is gradually sharpened because the faster the scanning speed per unit time, the longer the length of the cladding channel and the shorter the existence of the heat-affected zone, resulting in peak temperature curves of different curvatures. With the gradual increase in the spot radius, the highest temperature of the peak temperature curve gradually decreases because the heat flux per unit area decreases, resulting in a decrease in the peak temperature.

### 3.3. Analysis of Stress Field Results

A unidirectional coupling is used to simulate the laser cladding process. At the end of the temperature field simulation, the temperature load obtained from the temperature field is introduced into the stress field as the initial condition for the stress field analysis. The auto time step and large deflection were turned on, the fixed supports were set up on the left side of the part, and stress field calculations were performed. The thermal stress maximum simulation results are shown in Table 6, and the residual stress distribution is shown in Figure 8. (1)–(9) in Figure 8 correspond to groups 1–9 experiments in the orthogonal experimental protocol. The maximum values of thermal stress in Table 6 are consistent with the maximum values of thermal stress in Figure 8.

As observed in Figure 8, the maximum thermal stress in the clad layer is mainly concentrated at the metallurgical bond, because the matrix and powder melt to form a melt pool, due to the difference between the matrix and powder materials, after melting various elements are reorganized according to the affinity between the elements to form a new phase, resulting in the concentration of thermal stress. The variance and extreme difference analysis was performed on the maximum value of thermal stress in the molten layer, as shown in Table 7 and Table 8.

The relationship between laser power, spot radius and scanning speed on the thermal stress maximum was investigated by using the extreme variance analysis. As can be seen from Table 7 and Table 8, the laser power showed significance (F = 45.708, *p* = 0.021 < 0.05) on the maximum value of thermal stress, indicating the presence of a main effect and a differential relationship between the laser power and the maximum value of thermal stress. The spot radius and scanning speed did not show significance on the maximum value of thermal stress (F = 13.882, *p* = 0.067 > 0.05; F = 1.944, *p* = 0.340 > 0.05), indicating that the spot radius and scanning speed do not have a differential relationship on the maximum value of thermal stress. In summary, the degree of influence of the three factors on the thermal stress maximum is: laser power > spot radius > scanning speed.

Based on the results of the extreme variance analysis, numerical simulations of the optimal horizontal melting parameters were performed, and the equivalent residual stress cloud and residual stress distribution graphs were extracted as shown in Figure 9 and Figure 10. It can be seen from Figure 9 that as the laser cladding process continues, the residual stress distribution at the end of the cladding is larger than the residual stress distribution at the beginning of the cladding because the continuous input of laser energy during the cladding process leads to the accumulation of heat, which eventually affects the distribution of residual stress.

From Figure 10, it can be seen that there are two peak points in the thermal stress cycle curve at the sampling point. As the laser beam approaches the sampling point, the material around the melt pool expands, the point is subjected to pressure, and the first peak thermal stress point appears at the sampling point. When the laser beam is located directly above the sampling point, the material at the sampling point is heated and melted, at which point the stress at the sampling point is rapidly reduced to a trough. As the laser beam moves away from the sampling point, the temperature here decreases, the melt pool begins to solidify, the stress gradually increases, and a second peak thermal stress point appears at the sampling point. As the temperature gradually decreases to room temperature, the thermal stress gradually decreases and finally stabilizes at a value of 281 Mpa, the residual stress at the sampling point, which is consistent with the residual stress distribution cloud. The maximum value of thermal stress at the optimal melting parameters is 696 MPa, which is smaller than the minimum value of thermal stress of 796 MPa in the orthogonal experiment, verifying the validity of the extreme variance analysis method.

### 3.4. Analysis of Fluid Field Results

#### 3.4.1. Melting Pool Melting

The melt pool grows rapidly under the action of the continuous laser input. Figure 11 shows the model melt pool growth and diffusion process, with screenshots of the melt pool every 0.005 s until the melt pool grows and takes shape. Red represents air, blue represents 4140 alloy structural steel matrix, and between the red and blue is the solid–liquid coexistence interval. At 0.025 s, the laser beam has not yet reached the surface of the substrate; thus, the melt pool has not been formed. At 0.03 s, the laser just reaches the surface of the substrate, and due to the concentrated energy of the laser beam, it melts the metal just by touching the surface of the substrate, forming a fine molten pool. At 0.035–0.08 s, the melt pool is initially formed. At 0.085–0.14 s, the melt pool is gradually formed, and the phenomenon of unfused powder appears, which is due to the certain radius of the laser spot, and as the powder accumulates, the unfused powder gradually appears, showing the cross-sectional “sharp angle” distribution. At 0.145–0.2 s, the melt pool grows further and spreads because with the continuous input of laser energy, the heat reaches the melting point at the edge of the melt pool after accumulation, and the melt pool expands further, but due to the limited laser energy and loading time, the melt pool cannot expand indefinitely, and the melt pool profile is finally formed.

#### 3.4.2. Melt Pool Solidification

The images are intercepted every 0.01 s, and Figure 12 shows the melt pool solidification cloud at the moment of 0.21–0.56 s. The red part is air, blue is matrix, and the solid–liquid coexistence area is between red and blue. At 0.21–0.26 s, the shrinkage of the edge positions on both sides of the melt pool solidification is obvious because of the combined effect of heat conduction of the matrix and heat convection of the air, which leads to faster solidification at the edge of the melt pool, the liquid phase in the melt pool gradually starts to transform, and the solid–liquid coexistence area increases. At 0.27–0.32 s, while the edges of the melt pool shrink on both sides, the bottom of the melt pool also gradually begins to solidify, and the size of the melt pool changes more obviously. At 0.33–0.38 s, most of the liquid in the melt pool transforms into a solid–liquid coexistence state, and the melt pool is further reduced. At 0.39–0.44 s, all the liquid in the melt pool is transformed into a solid–liquid coexistence state, and the solidification is more obvious at both sides of the edge of the melt pool than at the bottom of the melt pool, with the melt pool showing a “funnel” shape. At 0.45–0.50 s, the melt pool gradually changes from a “funnel” shape to a “gourd” shape, which is due to the smaller size of the melt pool in the late solidification stage, and the different driving forces on both sides and at the bottom of the melt pool result in different solidification rates, finally showing a “gourd” shape. At 0.51–0.56 s, the melt pool solidification reaches the final stage, the “gourd” shape disappears, the melt pool is converted from solid–liquid coexistence to solid state, and the cladding layer is formed.

#### 3.4.3. Melt Pool Flow

Figure 13, Figure 14, Figure 15, Figure 16 and Figure 17 show the melt pool flow rate vector plot for the period 0.01 to 0.20 s. Due to the energy distribution characteristics of the Gaussian heat source, the middle is high, and the sides are low, and the shape of the melt pool at all times is characterized by a thick center and thin sides. At 0.01 s, the laser has not yet acted on the surface of the substrate; thus, neither the substrate nor the air has generated a flow rate. At 0.02–0.03 s, the laser acts on the surface of the substrate, but the action time is short, and only air convection is formed on the surface of the substrate, and the melt pool is not formed. At 0.04–0.07 s, with the continuous loading of the laser, the surface of the melt pool gradually formed, and the liquid flow rate inside the melt pool was gradually generated, but the flow rate inside the melt pool was very small, and the internal flow trend was not obvious. Thus, it was presumed that the solid–liquid coexistence state and the liquid metal were less inside the melt pool at this time. At 0.08 s, a larger flow rate of 0.005 m/s starts to appear in the melt pool at the center of the laser loading, which indicates that the liquid metal inside the melt pool increases and gradually forms inside the melt pool, driven by the driving force. At 0.09–0.11 s, with the loading of the laser, the melt pool is gradually formed, the melt pool enters the quasi-steady state, due to the symmetry of the laser, the flow velocity vector distribution is on the left and right sides, a vortex flowing from the middle to both sides is formed on each side near the edge, a small depression appears in the middle of the melt pool, and the melt pool formed is shallow and wide. At this time, the melt pool flow velocity increases from 0.005 to 0.01 mm/s. At 0.12–0.19 s, the melt pool gradually expands, the double vortex phenomenon inside the melt pool becomes more and more obvious, and the maximum flow velocity of the melt pool increases from 0.01 to 0.018 m/s. At 0.12 s, the vortex effect appears during air convection. Because of the symmetry of the laser, the vortex effect at the air convection is also symmetrical, forming two vortices flowing from the middle to both sides, because with the continuous input of laser energy, the air conducts heat conduction, hot air rises and cold air falls, eventually forming the air double vortex phenomenon. At 0.20 s, the melt pool is finally formed, and under the combined effect of surface tension and Marangoni effect, the double vortex phenomenon of the melt pool and air is most obvious at this time, and the flow velocity vector shows a tendency to flow from the bottom to top and from the center to all around with a maximum flow velocity of 0.02 m/s.

In summary, the internal flow state of the melt pool can be divided into four stages. The first stage of the melt pool did not produce internal flow velocity, and the second stage of the melt pool produced fine flow velocity. It is presumed that for the internal solid–liquid coexistence of the melt pool, the third stage of the melt pool reached quasi-steady state, producing a double vortex effect. The fourth stage of the melt pool double vortex phenomenon becomes gradually obvious, and the flow velocity gradually reached the maximum.

## 4. Conclusions

In this paper, laser cladding of 4140 alloy structural steel substrate and preparation of Inconel 718 alloy coating on its surface are carried out in the context of shaft parts repair, and multi-field coupled numerical simulation of the laser cladding process is carried out using ANSYS Workbench software. The conclusions are as follows.

(1) Since the temperature-dependent thermophysical properties of the matrix and powder materials are unknown, this paper uses the CALPHAD method to calculate the temperature-dependent thermophysical parameters of the matrix and powder and exports the parameters into ANSYS Workbench software for subsequent numerical simulations of laser cladding.

(2) The thermal stress field numerical simulation was carried out to analyze the thermal cycle curve and temperature distribution cross-section diagram in the temperature field and the residual stress distribution curve and residual stress cross-section diagram in the stress field The optimal laser cladding parameters were synthesized as laser power 1000 W, spot radius 1.5 mm and scanning speed 15 rad/s. The degree of influence of the three factors on the maximum value of thermal stress is: laser power > spot radius > scanning speed. The value of residual stress tends to be 281 Mpa at the optimal laser cladding parameters.

(3) Fluid field simulation was carried out under the optimal laser cladding parameters, and the melting pool melting, solidification and velocity vector clouds were extracted to analyze the inside of the melting pool. During the solidification process, the initial melt pool shape is wide and shallow, gradually changing to a “funnel” shape, then to a “gourd” shape, and finally, the solidification of the clad layer is completed. The internal flow state of the melt pool is divided into four stages. The first stage (0.01–0.03 s) does not generate a flow rate inside the melt pool. The second stage (0.04–0.07 s) produces a fine flow rate inside the melt pool, which is presumed to be a solid–liquid coexistence state inside the melt pool. In the third stage (0.08–0.11 s), the melt pool reaches a quasi-steady state, generating a double vortex effect, and the flow velocity increases from 0.005 to 0.01 m/s. The fourth stage of the melt pool (0.12–0.20 s) double vortex phenomenon is gradually obvious, and the flow velocity gradually reaches the maximum value of 0.02 m/s.

## Figures and Tables

**Figure 1 micromachines-14-00493-f001:**
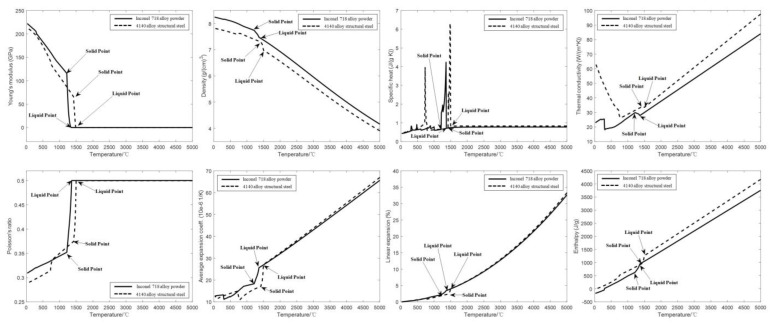
Thermophysical properties of substrate and powder.

**Figure 2 micromachines-14-00493-f002:**
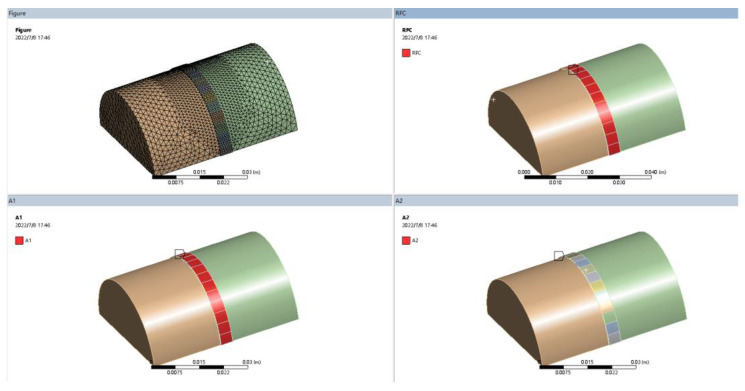
Naming results of each part and grid division figure.

**Figure 3 micromachines-14-00493-f003:**
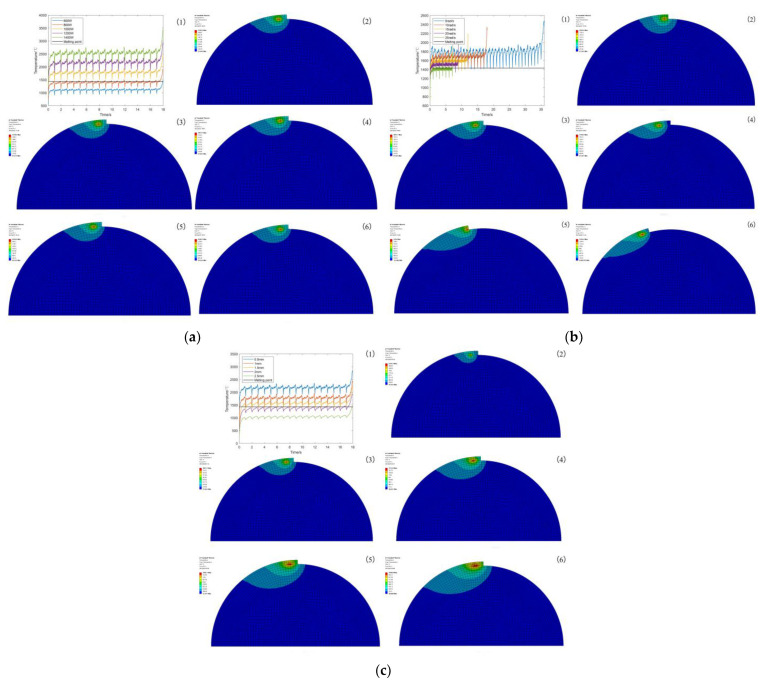
Simulation results of temperature field under different process parameters. (**a**) Different laser powers. (**b**) Different scanning speeds. (**c**) Different spot radius.

**Figure 4 micromachines-14-00493-f004:**
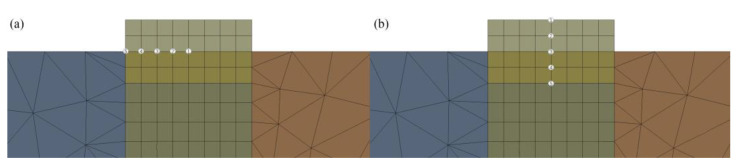
(**a**) *X*-axis sampling points and numbers. (**b**) *Y*-axis sampling points and numbers.

**Figure 5 micromachines-14-00493-f005:**
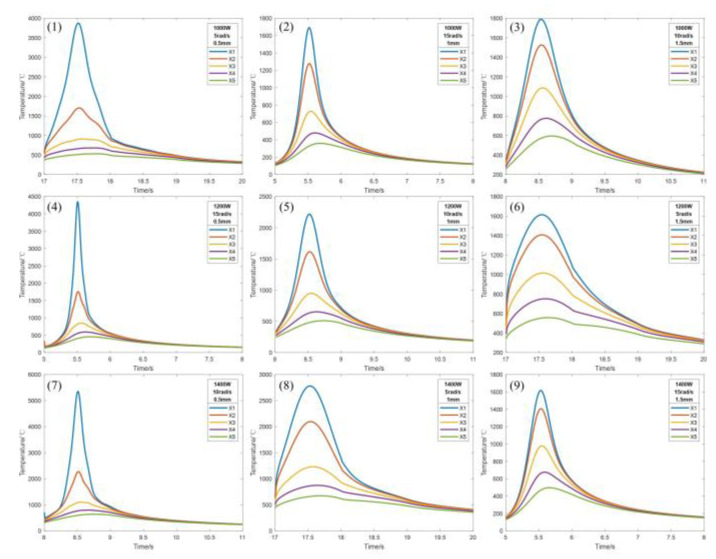
*X*-axis sampling point temperature variation curve.

**Figure 6 micromachines-14-00493-f006:**
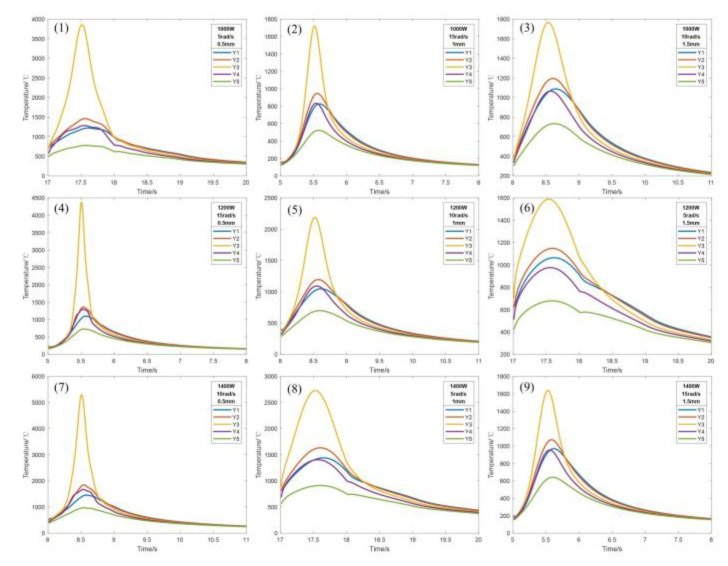
*Y*-axis sampling point temperature variation curve.

**Figure 7 micromachines-14-00493-f007:**
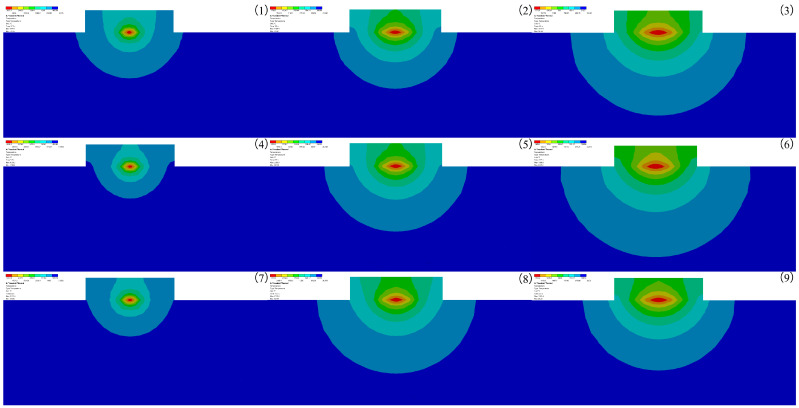
Contour plot of the temperature distribution of experimental groups 1–9.

**Figure 8 micromachines-14-00493-f008:**
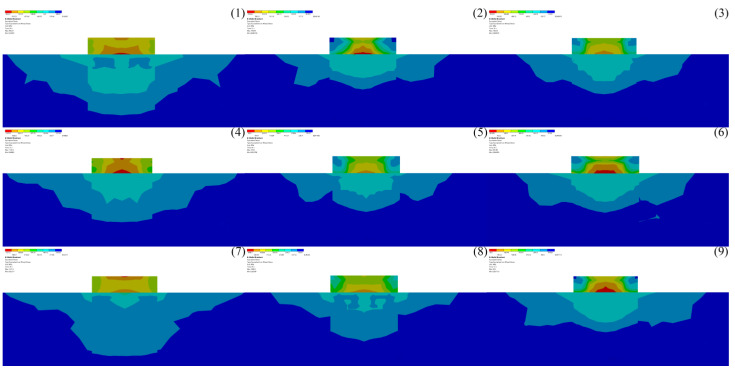
Von Mises stress distribution clouds of experimental groups 1–9.

**Figure 9 micromachines-14-00493-f009:**
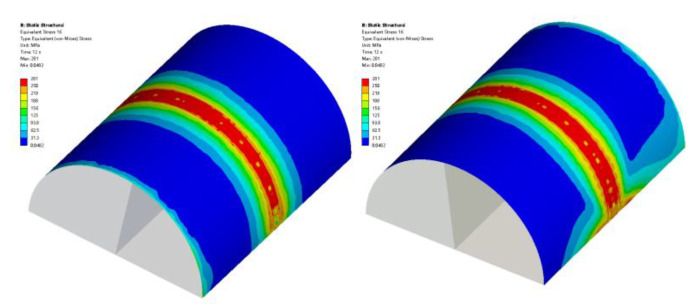
Residual stress distribution cloud.

**Figure 10 micromachines-14-00493-f010:**
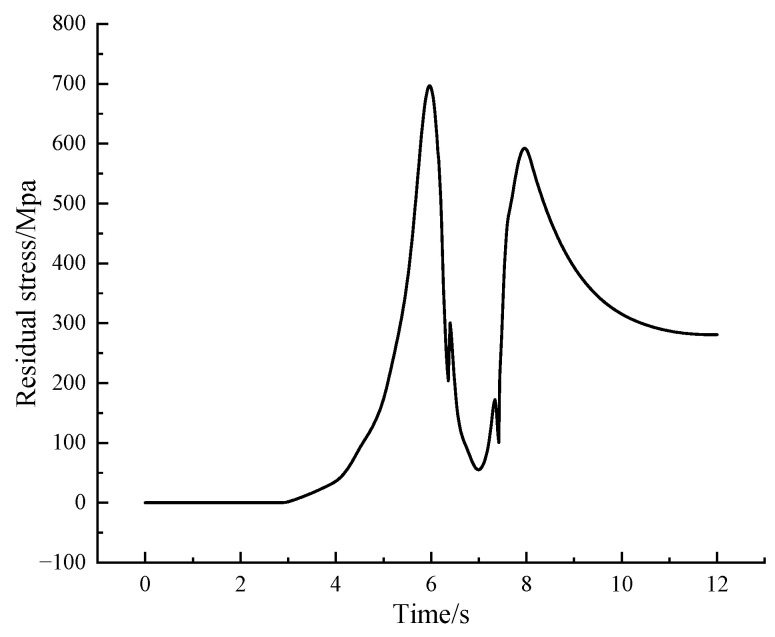
Residual stress distribution curve.

**Figure 11 micromachines-14-00493-f011:**
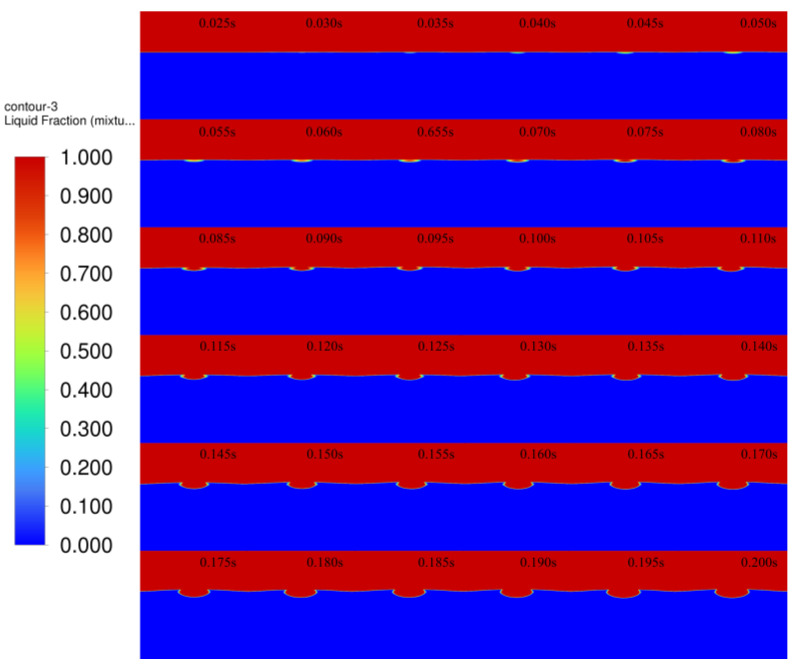
The 0.025–0.200 s melt pool melting cloud.

**Figure 12 micromachines-14-00493-f012:**
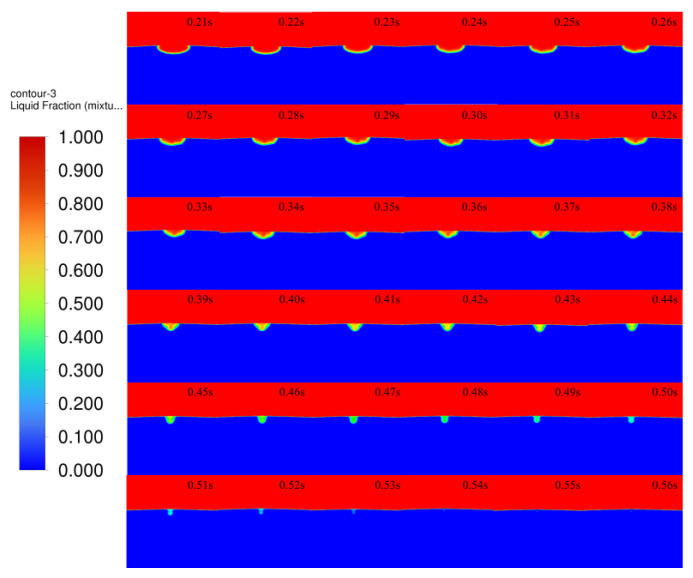
The 0.21–0.56 s melt pool solidification cloud.

**Figure 13 micromachines-14-00493-f013:**
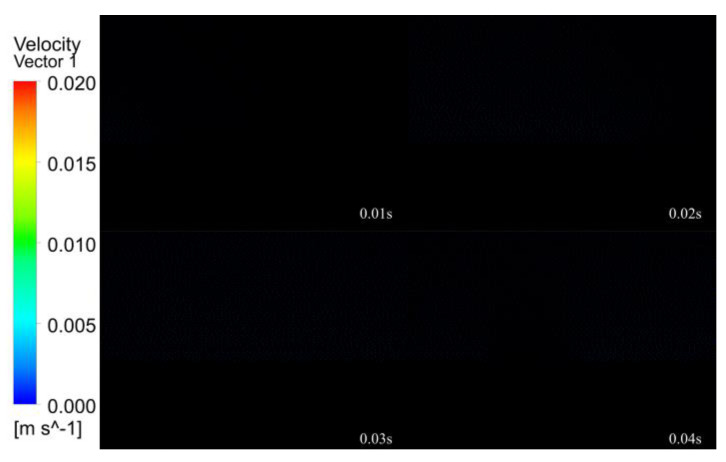
The 0.01–0.04 s melt pool flow rate vector plot.

**Figure 14 micromachines-14-00493-f014:**
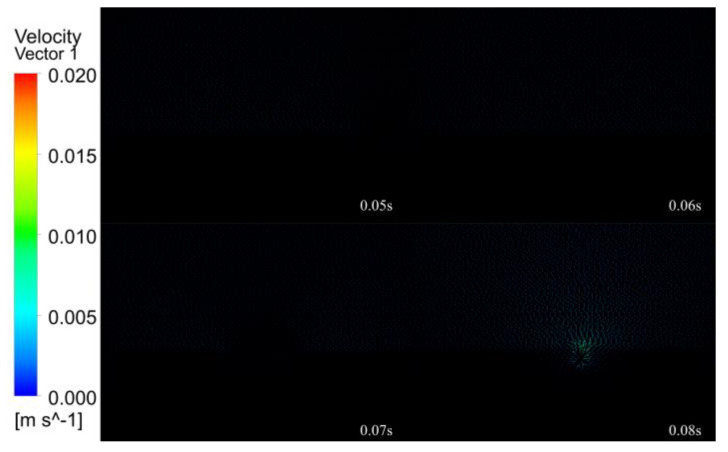
The 0.05–0.08 s melt pool flow rate vector plot.

**Figure 15 micromachines-14-00493-f015:**
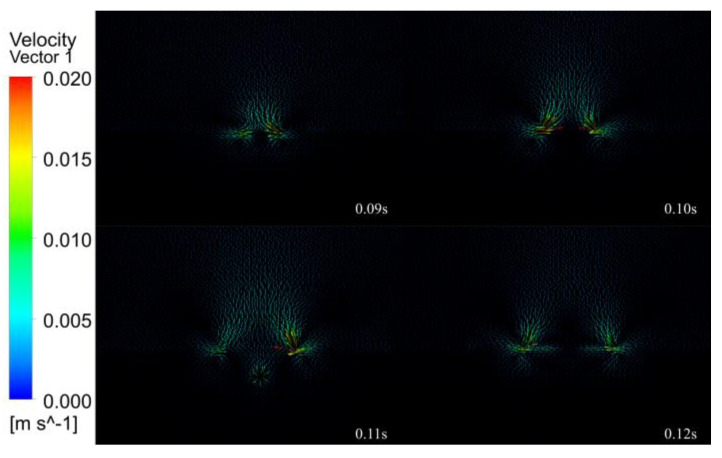
The 0.09–0.12 s melt pool flow rate vector plot.

**Figure 16 micromachines-14-00493-f016:**
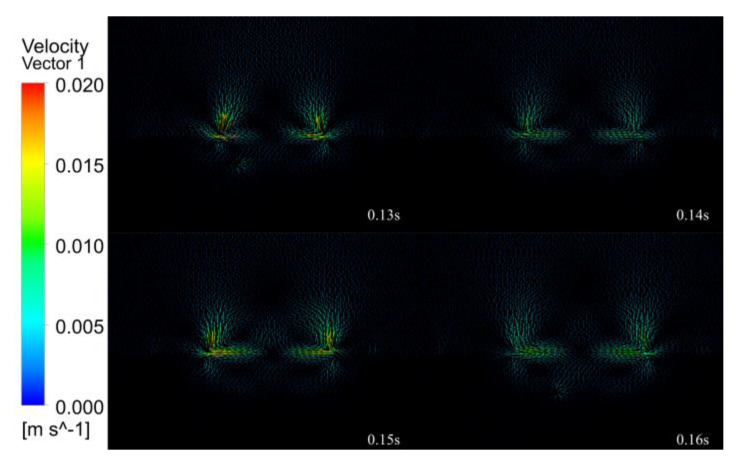
The 0.13–0.16 s melt pool flow rate vector plot.

**Figure 17 micromachines-14-00493-f017:**
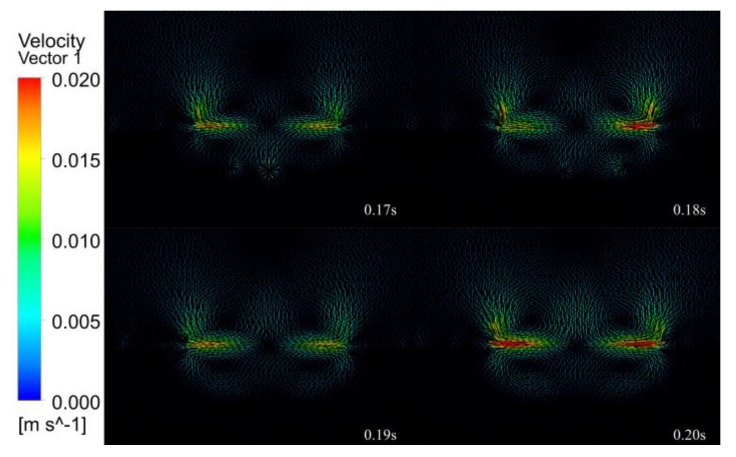
The 0.17–0.20 s melt pool flow rate vector plot.

**Table 1 micromachines-14-00493-t001:** Elemental composition of 4140 alloy structural steel.

Element	Cr	Mn	Mo	Ni	Si	C	P	S
Mass fraction/%	0.98	0.77	0.21	0.04	0.15	0.37	0.03	0.04

**Table 2 micromachines-14-00493-t002:** Inconel 718 alloy powder elemental composition.

Element	Al	Cr	Fe	Mo	Nb
Mass fraction/%	0.5	19.0	18.5	3.0	5.1

**Table 3 micromachines-14-00493-t003:** Experimental scheme of numerical simulation with different process parameters.

Fixed Factors	Variable Factors
Scanning speed 10 rad/s Spot radius 1 mm Remain unchanged	Laser power (W)
	600
	800
	1000
	1200
	1400
Laser power 1000 W Spot radius 1 mm Remain unchanged	Scanning speed (rad/s)
	5
	10
	15
	20
	25
Laser power 1000 W Scanning speed 10 rad/s Remain unchanged	Spot radius (mm)
	0.5
	1
	1.5
	2
	2.5

**Table 4 micromachines-14-00493-t004:** Peak temperature at different process parameters.

Fixed Factors	Variable Factors	Temperature
Scanning speed, spot radius certain Change laser power	600 W	1129.3 °C
	800 W	1418.6 °C
	1000 W	1811.1 °C
	1200 W	2203.9 °C
	1400 W	2583.1 °C
Laser power, spot radius certain Change scanning speed	5 rad/s	1953.6 °C
	10 rad/s	1811.1 °C
	15 rad/s	1709.9 °C
	20 rad/s	1454.0 °C
	25 rad/s	1456.2 °C
Laser power, scanning speed is certain Change spot radius	0.5 mm	2134.3 °C
	1 mm	1811.1 °C
	1.5 mm	1772.8 °C
	2 mm	1487.7 °C
	2.5 mm	1058.8 °C

**Table 5 micromachines-14-00493-t005:** Table of orthogonal experimental protocols.

No.	Laser Power (W)	Spot Radius (mm)	Scanning Speed (rad/s)
1	1000	0.5	5
2	1000	1	15
3	1000	1.5	10
4	1200	0.5	15
5	1200	1	10
6	1200	1.5	5
7	1400	0.5	10
8	1400	1	5
9	1400	1.5	15

**Table 6 micromachines-14-00493-t006:** Thermal stress maximum simulation results.

No.	Laser Power (W)	Spot Radius (mm)	Scanning Speed (rad/s)	Thermal Stress Maximum (Mpa)
1	1000	0.5	5	986
2	1000	1	15	796
3	1000	1.5	10	1033
4	1200	0.5	15	1147
5	1200	1	10	1074
6	1200	1.5	5	881
7	1400	0.5	10	1221
8	1400	1	5	1068
9	1400	1.5	15	853

**Table 7 micromachines-14-00493-t007:** Analysis of variance results for thermal stress maximum.

Source of Difference	Sum of Squares	df	Root Mean Square	F	P
Intercept	1,630,250	1	1,630,249.797	124.897	0.008 **
Laser power (W)	1,193,225	2	596,612.392	45.708	0.021 **
Spot radius (mm)	362,394.3	2	181,197.157	13.882	0.067
Scanning speed (rad/s)	50,754.89	2	25,377.444	1.944	0.34
Residual	26,105.56	2	13,052.778		
R^2^: 0.836	* *p* < 0.05	** *p* < 0.01			

**Table 8 micromachines-14-00493-t008:** Thermal stress maximum extreme difference analysis results.

Item	Level		Thermal Stress Maximum
K-value	Spot radius	0.5	3354
		1	2938
		1.5	2767
	Scanning speed	5	2935
		10	3328
		15	2796
	Laser power	1000	2815
		1200	3102
		1400	3142
K avg value	Spot radius	0.5	1118
		1	979.33
		1.5	922.33
	Scanning speed	5	978.33
		10	1109.33
		15	932
	Laser power	1000	938.33
		1200	1034
		1400	1047.33
Optimal level	1000	1.5	15
R	−109	−195.67	−177.33
Number of levels	3	3	3
Number of repetitions per level r	3	3	3

## Data Availability

Not applicable.

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
