# Peer review of "Numerical Simulation Study of Multi-Field Coupling for Laser Cladding of Shaft Parts"

_micromachines, 2023, doi:10.3390/mi14020493_

Round 1

Reviewer 1 Report

The manuscript entitled “micromachines-2198137-Cladding” dealing with AM has been reviewed. The paper has been nicely written but needs significant improvement. Please follow my comments.

1.      Add a short note about the results to the abstract.

2.      How authors reported “Table 1. Elemental composition of 4140 alloy structural steel”

3.      How authors selected the process parameters for the experimentation. This is important for repeating the test.

4.      The text has some typos. Please check them.

5.      Please provide a reference for the graphs in Figure 1.

6.      Cladding is a type of directed energy deposition (DED) and has usage in additive manufacturing. Add a short note to your introduction based on the usage of additive manufacturing and add the following four papers.

·       Comparison of properties at the interface of deposited IN625 and mixture of IN625 SS304L by laser directed energy deposition and SS304L substrate

·       Thermo-fluid flow behavior of the IN718 molten pool in the laser directed energy deposition process under magnetic field

·       Additive manufacturing of Ti-Al functionally graded material by laser based directed energy deposition

·       Sandwich structure printing of Ti-Ni-Ti by directed energy deposition

Author Response

Point 1: Add a short note about the results to the abstract.

Response 1: We have reworked the abstract content to include a short description of the results.

Point 2: How authors reported “Table 1. Elemental composition of 4140 alloy structural steel”.

Response 2: The elemental composition of the 4140 alloy structural steel in Table 1 was derived from the JMatPro software, and the relevant notes have been added in line 98 of the text.

Point 3: How authors selected the process parameters for the experimentation. This is important for repeating the test.

Response 3: The experimental parameters were selected based on the relevant references and the pre-preparation experiments, which have been added in the text in lines 179 to 185.

Point 4: The text has some typos. Please check them.

Response 4: We rechecked the content of the paper and corrected the typos.

Point 5: Please provide a reference for the graphs in Figure 1.

Response 5: Figure 1 is calculated by JMatPro software, exporting the calculation results and plotting them, which has been added in lines 106 to 109 of the paper.

Point 6: Cladding is a type of directed energy deposition (DED) and has usage in additive manufacturing. Add a short note to your introduction based on the usage of additive manufacturing and add the following four papers.

Response 6: The use of laser cladding in additive manufacturing has been added in the preamble section of the paper, lines 58 to 64, and the four recommended references are cited.

Reviewer 2 Report

The author has done a lot of work to study the laser cladding of 4140 alloy structural steel substrate. This work has a certain application prospect, but the work content needs to be further improved.

 Point 1: All the images that appear in the manuscript are very unclear. High quality figures are encouraged.

 Point 2: The summary of references (from line 46) should be integrated with the research to more reasonably illustrate the creativity of the thesis.

 Point 3: The studies referred to in lines 112 and 125 should cite the relevant references.

 Point 4: The experimental protocol designed in line 156 should be given a specific description and should preferably be presented in tabular form.

 Point 5: The "Simulation of temperature field under different process parameters" for line 164 should be placed in the Analysis of Results section.

 Point 6: The necessary explanations should be given for "group 1, 3, 4, 5, 6, 7, 8 and 9" in line 214, or the corresponding tables should be given.

 Point 7:  In the Conclusion section, only analytical results are given for the study of the melt pool and no clear conclusions are drawn.

Author Response

Point 1: All the images that appear in the manuscript are very unclear. High quality figures are encouraged.

Response 1: The image has been replaced with a clearer version and resized to the right size according to the typography.

Point 2: The summary of references (from line 46) should be integrated with the research to more reasonably illustrate the creativity of the thesis.

Response 2: A note has been added to lines 83 to 86 and 88 to 95 of the paper to combine the references with the content of the paper and to better illustrate the innovative nature of the paper.

Point 3: The studies referred to in lines 112 and 125 should cite the relevant references.

Response 3: According to the reviewer's comments, relevant references have been cited in lines 145, 154, 179, and 181 of the paper.

Point 4: The experimental protocol designed in line 156 should be given a specific description and should preferably be presented in tabular form.

Response 4: A table of experimental protocols has been added to line 189 of the paper, and related notes have been added to lines 179 to 185.

Point 5: The "Simulation of temperature field under different process parameters" for line 164 should be placed in the Analysis of Results section.

Response 5: Temperature field simulations for different process parameters have been placed in the results analysis section.

Point 6: The necessary explanations should be given for "group 1, 3, 4, 5, 6, 7, 8 and 9" in line 214, or the corresponding tables should be given.

Response 6: Experimental groups 1-9 are represented in Table 6, and to avoid repetition of the table, descriptions related to experimental groups 1-9 are added in rows 230 to 235, citing the relevant contents of Table 6.

Point 7: In the Conclusion section, only analytical results are given for the study of the melt pool and no clear conclusions are drawn.

Response 7: Relevant conclusions were added to the melt pool analysis section and the conclusion section, where the internal flow rate of the melt pool was re-analyzed in lines 402 to 421, and in lines 422 to 427, and in lines 450 to 457, conclusions were added.

Round 2

Reviewer 1 Report

The paper is ready to publish.

Author Response

We sincerely thank our reviewers for taking the time to review our paper and your valuable suggestions have helped us to improve the quality of our paper.

Reviewer 2 Report

Point 1: I cannot identify if you have changed the figures because the figures given in your revision are still not clear!

Point 2: The description of the article in lines 85-86 should be placed somewhere after the conclusion of the literature review in lines 87-88. You should make reasonable revision.

Point 3: You mentioned Table 6 in line 216. Therefore, you should have explained the contents of Table 6 here or earlier in the article rather than later.

Author Response

Many thanks to the reviewers for their valuable comments, which are very helpful in improving the quality of our articles.

Response to the comments one by one

Point 1: I cannot identify if you have changed the figures because the figures given in your revision are still not clear!

Response 1: We thank the reviewers for their valuable comments and we have reworked the figures to make changes. The figures in lines 108-109 and 135-136 have been enlarged, and the figures in lines 195-199, 237-238, 293-294, and 337 have been replaced with high-resolution figures and enlarged, and the modified figure captions have been marked in red in the paper.

Point 2: The description of the article in lines 85-86 should be placed somewhere after the conclusion of the literature review in lines 87-88. You should make reasonable revision.

Response 2: The description in lines 85-86 of the article has been combined with the description in lines 87-88, and the re-narrated words are in lines 84-93 of the paper, which have been marked in red in the paper.

Point 3: You mentioned Table 6 in line 216. Therefore, you should have explained the contents of Table 6 here or earlier in the article rather than later.

Response 3: A table of orthogonal experimental protocols has been added to line 218 of the paper, placing the explanation of experimental groups 1-9 on lines 212-218, which have been marked in red.
